# Thermal Ablation Experiments of Carbon Phenolic and SiC-Coated Carbon Composite Materials Using a High-Velocity Oxygen-Fuel Torch

**DOI:** 10.3390/ma16051895

**Published:** 2023-02-24

**Authors:** Rajesh Kumar Chinnaraj, Young Chan Kim, Seong Man Choi

**Affiliations:** Department of Aerospace Engineering, Jeonbuk National University, Jeonju 54896, Republic of Korea

**Keywords:** TPS, spacecraft heat shield, HVOF, ablation, carbon phenolic, C–C, SiC

## Abstract

For future spacecraft TPS (heat shield) applications, ablation experiments of carbon phenolic material specimens with two lamination angles (0° and 30°) and two specially designed SiC-coated carbon–carbon composite specimens (with either cork or graphite base) were conducted using an HVOF material ablation test facility. The heat flux test conditions ranged from 3.25 to 11.5 MW/m^2^, corresponding to an interplanetary sample return re-entry heat flux trajectory. A two-color pyrometer, an IR camera, and thermocouples (at three internal locations) were used to measure the specimen temperature responses. At the 11.5 MW/m^2^ heat flux test condition, the 30° carbon phenolic specimen’s maximum surface temperature value is approximately 2327 K, which is approximately 250 K higher than the corresponding value of the SiC-coated specimen with a graphite base. The 30° carbon phenolic specimen’s recession value is approximately 44-fold greater, and the internal temperature values are approximately 1.5-fold lower than the corresponding values of the SiC-coated specimen with a graphite base. This indicates that increased surface ablation and a higher surface temperature relatively reduced heat transfer to the 30° carbon phenolic specimen’s interior, leading to lower internal temperature values compared to those of the SiC-coated specimen with a graphite base. During the tests, a phenomenon of periodic explosions occurred on the 0° carbon phenolic specimen surfaces. The 30° carbon phenolic material is considered more suitable for TPS applications due to its lower internal temperatures, as well as the absence of abnormal material behavior as observed in the 0° carbon phenolic material.

## 1. Introduction

Thermal protection systems (TPS) or heat shields are essential for any earth return space mission. During atmospheric re-entry, atmospheric drag and aerodynamic heating cause the air around the spacecraft to undergo dissociation and ionization, thereby a high thermal plasma flow is generated around the re-entering spacecraft. This thermal plasma flow exerts extreme heat flux conditions on the spacecraft. So, to protect the spacecraft, a heat shield or a TPS is used. Hence, for earth return space missions, the TPS is a single-point-of-failure (SPOF) system [1], meaning that if the TPS fails, the entire mission fails, leading to complete destruction of the spacecraft. Thereby, careful selection and validation of TPS-candidate materials is required for development of an effective TPS to withstand extreme heat loads, and which must also maintain spacecraft structural integrity during re-entry. 

The TPS/TPS material can be broadly classified into two types, based on how they react to the extreme thermal flow during re-entry: (1) re-usable, non-ablative and non-charring; (2) non-reusable and ablative, including charring and non-charring subtypes [2]. 

An example of a re-usable TPS is reinforced carbon–carbon (RCC) used for US space shuttles [3]. Re-usable TPS materials undergo no physical or chemical changes after the re-entry process and, in general, can only withstand low heat flux conditions that occur during low earth orbit re-entries. Typically, a reusable TPS operates by re-radiating a considerable amount of the incident heat and absorbing the remaining heat. Hence, sometimes, a reusable TPS is also known as a heat-soak TPS. 

The non-reusable TPS materials, especially charring ablators, are the most commonly used since they are very suitable for high earth orbit and interplanetary space missions. All NASA interplanetary spacecraft have used ablative TPS materials [1], including the recently developed Orion spacecrafts [4]. Charring ablators mostly contain an organic resin such as phenolic combined with re-enforcement phases such as carbon [5]. During the carbon phenolic ablation, pyrolysis or the thermal degradation process of the phenolic resin an endothermic process takes place, which absorbs a fraction of the heat. Additionally, the blockage effect caused by pyrolysis gases over the ablating surface decreases a fraction of the heat transferred to the surface. Hence, phenolic resin-based charring ablators are more favored than other types of TPS materials. AVCOAT (Apollo), PICA (Stardust), and MC-CFRP (Hayabusa) are some examples of the successful phenolic resin-based ablators developed and used over the years.

In this study, two types of carbon phenolic ablators were fabricated using identical materials and methods, but with different lamination angles (0° and 30°). The ablation recession rate is known to be affected by fiber orientation in C–C materials [6] and similar can be expected in carbon phenolic ablators with different lamination angles.

Two specially designed SiC-coated C–C specimens with two different bases were also fabricated. SiC is a hard, refractory, semiconductor material with a wide range of industrial applications. It melts incongruently at approximately 2840 K, meaning the Si fraction liquefies first, leaving the C fraction as a solid [7]. SiC is typically used in the form of ceramic matrix composites (CMCs) for high-temperature aerospace applications. The SiC-based CMCs are used in gas turbine engine exhaust cones, casing, struts, and primary and secondary nozzle flaps [8]. SiC is also studied as a coating layer to improve high-temperature oxidation resistance of C–C material substrates [9,10,11]. SiC materials, due to their favorable neutronic properties, are useful for nuclear applications [12] and are being studied as temperature monitors for nuclear reactors [13]. Other non-high-temperature applications of SiC include nanowires for LED applications [14,15]; and in optical instruments for terrestrial and space applications [16]. New methods for graphene synthesis [17,18] can be explored to improve ablative material properties in the future. 

Each SiC-coated C–C specimen was fabricated with a YSZ (Yttria-stabilized zirconia) intermediary layer; for more details on specimen fabrication, see Section 2. This is the first time that specimens made from this material combination have been tested for TPS (heat shield) applications and compared with carbon phenolic ablators. The ablation response data of the SiC-coated specimens will be useful for research and development of new types of TPS materials. 

The investigations of the specimens were carried out by exposing the specimens to flows generated using the JBNU’s HVOF material ablation test facility. Originally designed for powder coating, the JBNU’s HVOF facility has been adapted for preliminary investigation of ablative materials [19] and designed to provide an easy-to-use alternative for JBNU’s 0.4 MW supersonic arc-jet plasma wind tunnel [20]. The JBNU’s HVOF material ablation test facility is one of its kind; only three other HVOF torch-based material test facilities have been reported so far [21,22,23,24]. In particular, the authors of [24] used the only other test facility reported for studying material ablation, while others [21,22,23] used test facilities mostly used for material fatigue testing.

In this study, the specimens were subjected to heat flux conditions consistent with an interplanetary re-entry heat flux trajectory. The data obtained from this study will be used to further develop and improve TPS (heat shield) materials for a future Korean interplanetary earth return spacecraft. The mass loss and recession data will help to predict the structural integrity of the heat shield; the internal temperature response will help to determine the heat shield thickness, as the heat shield back-face temperature should be tolerable to the internal structure of the spacecraft during re-entry. This work can serve as a useful reference for designing, low-cost testing and characterizing materials for spacecraft TPS applications. 

## 2. Materials and Methods

### 2.1. Specimens

Figure 1 shows the fabrication process of the carbon phenolic specimens (final density ≈ 1340 kg/m^3^). First, a bulk carbon phenolic block was fabricated by combining a rayon-based carbon fabric with a resol phenolic resin, followed by vacuuming and hydroclave processes. The desired lamination angles were obtained by cutting the processed bulk carbon phenolic block using a hole cutter at certain angles. 

Figure 2 shows the manufacturing process of the SiC-coated C–C specimens. Each SiC-coated specimen had two sections, (1) a circular coin-shaped top section: SiC-coated C–C sample with bottom surface coated with YSZ, and (2) a T-shaped bottom section: either made from cork or graphite felt. First, bulk C–C was fabricated from carbon fiber preform, then the bulk C–C was machined into two circular coin-shaped C–C samples (diameter: 30 mm and thickness: 10 mm). Subsequently, the C–C samples were coated with SiC using a CVR (chemical vapor reaction) process. Next, bottom surfaces of the SiC-coated samples, i.e., the surfaces opposite to the flow-facing surfaces, were coated with YSZ using the APS (atmospheric plasma spray) process. YSZ coating thickness was 1000 µm. To be clear, in a coated SiC-coated C–C top section sample, the bottom surface refers to the YSZ coated side, and the top surface refers to the opposite side which was exposed to the HVOF flow during the test. Using a carbon adhesive, one coated sample was attached to a T-shaped cork bottom section and another coated sample was attached to a T-shaped graphite bottom section. The T-shaped bottom sections had a lower stem part (diameter: 18 mm, length: 35 mm), and an upper circular part (thickness: 5 mm and diameter: 30 mm). 

Figure 3 shows the specimen dimensions. The specimens were machined with three slots, enabling internal temperature measurements at three locations, i.e., 10, 20, and 30 mm from the stagnation point. The thermocouple slots were 1.6 mm in diameter each, placed at 3.5 mm from the specimen center and at an angle of 120° from each other.

During the HVOF tests, the specimens were flush mounted in a specimen holder made of graphite (see Figure 4) to minimize any heat transfer from lateral directions.

### 2.2. Experimental Setup

The torch of the JBNU’s HVOF material ablation test facility has a nozzle exit diameter of 10.84 mm and can produce high-velocity and high-temperature flows, capable of simulating atmospheric re-entry heat flux conditions. Figure 5 shows a photograph of the experimental setup.

A two-color pyrometer (IMPAC series ISQ 5 model from LumaSense Technologies, Santa Clara, CA, USA with a measurement range from 1073.15 K to 3073.15 K) was used to measure the specimen stagnation point temperatures during the tests. An infrared (IR) camera (FLIR A655SC, theoretically, can measure up to 2273.15 K and measurements higher than 2273.15 K are also possible under certain conditions) was used to capture IR images of the specimen flow-facing surfaces during the tests. From IR camera images, specimen stagnation point temperatures were estimated and compared with the pyrometer data. The IR camera emissivity was set arbitrarily during the tests and later, during data analysis, the emissivity values were adjusted, until the IR camera temperature data matched with the pyrometer values. A camcorder and a high-speed camera were used to observe and record the specimen reactions during the tests. 

During the tests, three K-type thermocouples were used to measure the specimen internal temperatures. The National Instruments NI cRIO-9054 (a real-time embedded industrial controller) with a thermocouple input module NI 9212 was used as the thermocouple data acquisition system. Figure 6 shows how the three thermocouples were inserted into a specimen through the specimen holder before a test.

Under a set of operating conditions, the HVOF flow’s heat flux varies inversely with the distance from the nozzle exit. Prior to the specimen tests, a water-cooled Gardon gauge was used to measure the HVOF flow’s stagnation point cold wall heat flux in the flow’s axial direction. The specimens were tested under the same set of operating conditions as the Gardon gauge experiments. For more details on the HVOF material ablation test facility and Gardon gauge dimensions, see [19,25]. 

Table 1 lists the tested specimens along with heat flux test conditions and test durations. Each carbon phenolic specimen was tested at one heat flux condition, whereas each SiC-coated C–C specimen was tested sequentially in three steps (i.e., three heat flux test conditions), first step: at 3.25 MW/m^2^ for 30 s, second step: at 6 MW/m^2^ for 40 s, and the third step: at 11.5 MW/m^2^ for 30 s. In between steps, there was significant time-gap/cool-off period, as it was needed to re-align optical devices such as the pyrometer and camcorder due to change in specimen position with respect to the nozzle exit. After-test SEM images of selected specimens were obtained using a high-resolution electron microscope (SU8030) from Hitachi, Tokyo, Japan. Specimen 3-D surface measurements were taken using a 3-D optical/non-contact VR-5200 measurement system from Keyence, Osaka, Japan. Specimen 3-D measurements were taken before and after tests, to study the specimen exposed surface morphology changes caused by the ablation.

## 3. Results and Discussion

Figure 7 shows the temperature responses of the 0° carbon phenolic specimens (CP (0°)-1 and CP (0°)-3) during the tests. The reactions of the 0° carbon phenolic specimens to the tests were very eventful. The 0° carbon phenolic specimens’ exposed surfaces exploded periodically during the tests. During the explosions, the laminate layers were expelled away from the specimen, almost in a layer-by-layer fashion; this is more pronounced in the CP (0°)-3 specimen, as evident from its after-test image. In Figure 7, the fluctuations observed in the stagnation point temperature data (both pyrometer and IR camera) correspond to the observed periodic explosion phenomenon. So far, this type of material reaction has not been reported yet; the main contributing factor to the periodic explosion phenomenon seems to be the laminate’s orientation angle (i.e., the lamination angle), as this phenomenon was not observed in the 30° carbon phenolic specimen tests. The intensity of the surface periodic explosions increased with increase in the heat flux, as evident from the increased stagnation point temperature fluctuations observed for the CP (0°)-3 specimen. During the CP (0°)-3 test, the thermocouple @ 10 mm failed to measure for unknown reasons.

Figure 8 shows an explosion event occurred during the CP (0°)-3 specimen test.

Figure 9 shows the before and after test images of the specimens CP (0°)-1 and CP (0°)-3. Figure 10 shows the before and after tests exposed surface morphologies of the specimens CP (0°)-1 and CP (0°)-3; taken using the 3-D optical/non-contact measurement system (i.e., the VR-5200 measurement system).

Figure 11 shows the temperature responses of the specimens CP (30°)-1 and CP (30°)-2 and Figure 12 shows the temperature responses of the specimens CP (30°)-3 and CP (30°)-4. The stagnation point temperature trends seen in Figure 11 and Figure 12 are stable compared to the 0° carbon phenolic specimens, as there was no explosion-induced expulsion of mass that occurred on the 30° specimens’ exposed surfaces. On comparing the CP (0°)-1 and CP (30°)-1 specimens, which were tested at identical heat flux test conditions, the internal temperatures were lower for the CP (0°)-1 specimen. The same is true for CP (0°)-3 and CP (30°)-2 specimens, which were also tested at identical heat flux test conditions; internal temperatures were lower for the CP (0°)-3 specimen when compared to the CP (30°)-2 specimen’s corresponding internal temperatures. The additional mass expulsion occurred during the periodic surface explosions may be one of the reasons for the lower internal temperatures observed in the 0° carbon phenolic specimens.

Figure 13 and Figure 14 show the before and after test images of the 30° carbon phenolic specimens. Figure 15 and Figure 16 show the before and after tests exposed surface morphologies of the 30° carbon phenolic specimens. The ablation process was different for the 30° carbon phenolic specimens compared to the 0° carbon phenolic specimens. For the 30° carbon phenolic specimens; the recession/material loss was approximately confined to the area around the test flow–material interaction zone. On the contrary, for the 0° carbon phenolic specimens, due to periodic explosions and layer-by-layer removal of laminate layers, the material loss occurred throughout the surface perpendicular to the test flow direction. This indicates that the test flow gas entered ablated surface holes or gaps of the 0° carbon phenolic specimens and propagated into the inter-laminate space between ablated and unablated layers. This process generated a significant amount of pressure in the inter-laminate space as the movement of penetrated test gas restricted by the unablated laminates being perpendicular to the test flow’s direction. This pressure led to layer-by-layer explosion in the direction perpendicular to the test flow, as seen in Figure 9. However, in the case of the 30° carbon phenolic specimens, since the laminates were obliquely cut, prevented the test gas from building pressure in the inter-laminate space. This happened as the pressure escaped through the slanted laminate surface.

Figure 17 and Figure 18 show camcorder and high-speed camera images of the CP (30°)-4 specimen test, respectively. No periodic explosion was observed; however, it was found in the high-speed camera images that fine fragments were intermittently ejected due to ablation.

Figure 19 shows IR images of the CP (30°)-4 specimen test (emissivity = 0.87). The IR images show that the CP (30°)-4 specimen’s surface temperature reached approximately 2100 K at ~1 s, whereas the maximum surface temperature measured by the IR camera for the CP (30°)-4 specimen test is 2344.60 K.

Figure 20 shows the temperature responses of the specimens SiC/cork and SiC/graphite. During the first test, i.e., at 3.25 MW/m^2^ for 30 s, the cork section of the SiC/cork charred near the end of the test and the cork section became separated from the SiC-coated C–C section at the carbon adhesive layer. For the SiC/cork specimen, further tests at higher heat flux conditions, i.e., at 6 MW/m^2^ for 40 s and 11.5 MW/m^2^ for 30 s were carried out using only the top SiC-coated C–C section, hence Figure 20 shows only the specimen stagnation point temperatures and not internal temperatures at these test conditions. No such charring event occurred for the SiC/graphite, and the specimen was structurally intact during all the three tests. In the 3.25 MW/m^2^ test, SiC/cork specimen’s internal temperatures were approximately 2- to 3-fold lower than the SiC/graphite specimen’s internal temperature, although both specimens’ stagnation point temperatures were similar. This is due to the lower thermal conductivity of the cork material.

Compared to the carbon phenolic specimens, the performances of the SiC-coated C–C specimens’ exposed surfaces to the tests were truly outstanding, even though both SiC-coated C–C specimens were tested at multiple heat flux test conditions for a total duration of 100 s per specimen. Figure 21 and Figure 22 show the before and after test images of the SiC-coated C–C specimens. Figure 21 shows no remarkable changes due to ablation, apart from changes in surface color and minor scratches on the exposed surfaces. Figure 21 indicates that the exposed surfaces of the SiC-coated C–C specimen were very resilient, even at the higher heat flux condition of 11.5 MW/m^2^. Figure 22 shows the charred cork section of the SiC/cork specimen and shows a clean and complete separation of the lower charred section from the top SiC-coated C–C section. Figure 22 also shows that SiC/graphite specimen was completely intact, even after the final 11.5 MW/m^2^ test. Figure 23 shows the SiC-coated C–C specimens’ exposed surface morphology changes. The 3-D surface measurements shown in Figure 23 confirm the visual observations made in Figure 21 that exposed surface changes occurred on the SiC-coated C–C specimens are almost non-existent. 

Figure 24 shows the test-wise maximum stagnation point temperatures measured by the pyrometer. As expected, the maximum stagnation point temperatures varied proportionally with the heat flux test conditions. The maximum stagnation temperatures SiC-coated C–C specimens were significantly lower than those of carbon phenolic specimens at identical heat flux test conditions. Figure 24 shows that the maximum stagnation point temperature trend shown by the 0° carbon phenolic specimens is steeper than that of the 30° carbon phenolic specimens, which may be due to the periodic explosions occurred on the 0° carbon phenolic specimens’ exposed surfaces. Figure 24 also, shows that the maximum stagnation point temperature trend shown by the 30° carbon phenolic specimens starts to level off as the heat flux increases and the maximum stagnation point temperature values of the SiC-coated C–C specimens are almost identical. 

For comparison, test-wise internal temperatures at 30 s from the test start are plotted and shown as Figure 25 for the carbon phenolic specimens, and as Figure 26 for the SiC-coated C–C specimens. As noted earlier, the corresponding internal temperatures of 0° carbon phenolic specimens were lower than those of 30° carbon phenolic specimens, this may be due to change in thermal conductivity because of different lamination angles. Additionally, the observed periodic explosion might have also reduced internal heat conduction. In Figure 25, at the 3.25 MW/m^2^ heat flux test condition, the internal temperature of the CP (0°)-1 specimen at 10 mm (from the stagnation point) is similar to the CP (30°)-1 specimen’s internal temperature at 20 mm and the CP (0°)-1 specimen’s internal temperatures at 20 and 30 mm are less than the CP (30°)-1 specimen’s internal temperature at 30 mm. Additionally, in Figure 25, at the 6 MW/m^2^ heat flux test condition, the CP (0°)-3 specimen’s internal temperatures at 20 mm is similar to the CP (30°)-2 specimen’s internal temperatures at 30 mm. In Figure 25, the 30° carbon phenolic specimens’ internal temperature trends at 20 and 30 mm are almost horizontally straight and parallel to each other, whereas the CP (30°)-3 specimen’s (at 9 MW/m^2^) internal temperature is slightly elevated at 10 mm, compared to other 30° carbon phenolic specimens. 

At the 3.25 MW/m^2^ heat flux test condition, the internal temperatures of the SiC/cork specimen were lowest. Though, compared to carbon phenolic specimens, test condition-wise stagnation point temperatures of the SiC-coated C–C specimens were the lowest, yet it did not translate to lower internal temperatures for the SiC/graphite specimen, the internal temperatures of the SiC/graphite specimen were the highest among the specimens. Figure 26 shows that the internal temperature trends of the SiC/graphite specimen at 20 and 30 mm are decreasing with increasing heat flux test conditions and are parallel to each other; whereas the 10 mm internal temperature trend of the SiC/graphite specimen is increasing with increasing heat flux test conditions. 

Figure 27 shows the test-wise specimen exposed surface emissivities estimated by matching the IR camera data with the pyrometer data. Figure 27 shows that the emissivity increases with heat flux test conditions, except for the CP (30°)-3 specimen test. Figure 27 also shows that the SiC/cork specimen’s estimated emissivity at the 11.5 MW/m^2^ test condition does not follow the general trend shown by estimated emissivities for other SiC-coated specimen tests. Ignoring the CP (30°)-3 specimen’s estimated emissivity as an outlier, the overall emissivity for the carbon phenolic specimens is 0.85 ± 0.02. Similarly, for the SiC-coated C–C specimens, ignoring the SiC/cork specimen’s test at 11.5 MW/m^2^ as an outlier, the overall emissivity is 0.87 ± 0.01.

Table 2 shows specimen-wise mass loss rate in g/min and recession rate in mm/min. Figure 28 shows the mass loss rate of surface recession vs. heat flux plot for the carbon phenolic specimens. The mass loss rate of surface recession was estimated using the Equation (1) [26]. The trend shown by the 0° carbon phenolic specimens is extremely steep compared to the 30° carbon phenolic specimens. Equation (2) shows the linear fit of the 30° carbon phenolic specimens’ mass loss rate of surface recession. Since the recessions shown by the SiC-coated C–C specimens were insignificantly minimal, the SiC-coated C–C specimens recession values are not plotted in Figure 28. In Figure 28, the trend of the mass loss rates of surface recession of the 30° carbon phenolic specimens is closer to its linear fit (which rises gradually compared to the steep rise of that of the 0° carbon phenolic specimens); this indicates that the mass loss rate of surface recession values exhibited by the 30° carbon phenolic specimens are normal and acceptable. Additionally, this linear fit (in Figure 28, Equation (2)) is valid only for heat flux test conditions ranging from 3.25 to 11.5 MW/m^2^. A linear or polynomial extrapolation is considered necessary for heat flux conditions other than the test conditions presented in this study.
(1)$$mass\;loss\;rate\;of\;surface\;recession\;\left[g\;m^{-2}s^{-1}\right]=\frac{Recession\;\left[m\right]}{Test\;duration\;\left[s\right]}\times specimen\;density\;\left[\frac{g}{m^{3}}\right]$$
(2)$$mass\;loss\;rate\;of\;surface\;recession\;of\;30{}^{\circ}\;\mathrm{carbon}\;\mathrm{phenolic}\;\mathrm{specimen}\left[g\;m^{-2}s^{-1}\right]\;=63.50\times heat\;flux\;\left[\frac{MW}{m^{2}}\right]-10.90$$

Figure 29 shows SEM images of exposed surfaces of the specimens CP (0°)-1, CP (30°)-4 and SiC/cork at 100×, 500×, and 5000× magnification levels. The exposed surface SEM images from the carbon phenolic specimens are quite similar and at 100× magnification level, carbon fibers damaged due to ablation are visible on the surfaces. The 500× and 5000× magnification levels show that these ablated carbon fibers have a sharply tipped needle-like shape. At the 5000× magnification level, the exposed surface SEM image of the SiC/cork specimen show the formation of a fused/partially fused SiO_2_ (silica) layer due to oxidation [19]. Figure 30 shows the cross-sectional SEM images of the specimens CP (0°)-1, CP (30°)-4 and SiC/cork at 500× magnification level. Prior to cross-sectional SEM, each specimen was cut cross-sectionally along the stagnation point and enclosed in an epoxy resin case. As observed in Figure 29 and Figure 30 also shows that the structures of ablated carbon fibers are similar for the carbon phenolic specimens. The cross-sectional SEM of the SiC/cork specimen shows the silica layer on top of the unablated base C–C. The minimal mass loss and recession rate observed in each SiC-coated C–C specimen were due to the formation of the fused/partially fused silica layer. The formed silica layers prevented further infiltration of the test gas into the SiC C–C coated specimens and shielded the inner C–C substrates from ablation. During the 11.5 MW/m^2^ tests, the formed silica layers were in molten state, as the specimen surface temperatures rose above 2000 K (the melting and boiling points of silica in an amorphous solid form are 1986.15 K and 3223.15 K, respectively [27]). Even in the molten state, the silica layer showed good adherence to the specimen surfaces, withstanding high dynamic pressure of the test flow.

The overall test results indicate that the 0° carbon phenolic material is not suitable for TPS application due to the observed periodic explosion phenomenon. The SiC-coated C–C specimens exhibited exemplary resistance to ablation despite having been subjected to extreme test conditions compared to the carbon phenolic specimens. However, in the case of SiC/Cork specimen, charring of the cork section and in the case of the SiC/graphite specimen, higher internal temperatures (>500 K, at the end of test), make them also not suitable for interplanetary sample return TPS applications. Taking all factors into account, 30° carbon phenolic material is more suitable for TPS application among other material/specimens tested in this study. Figure 31 shows thermal diffusivity (α) and Cp (specific heat at constant pressure) of the 30° carbon phenolic material measured up to 773.15 K (i.e., 500 °C) using an LFA 467 Hyper Flash apparatus from NETZSCH, Germany. Thermal conductivity (κ) was determined using Equation (3) (where ρ is material density) and plotted in Figure 31.
(3)$$\kappa =\rho \times \alpha \times C_{p}$$

Milos et al. [28] manufactured a dual-layer woven carbon phenolic material from a 3-D woven substrate and tested at various heat flux test conditions using plasma wind tunnels, including at 3.25 MW/m^2^, i.e., the same heat flux test condition as the CP (30°)-1 specimen. Here, in Figure 32, Milos et al. specimen’s temperature responses at 3.25 MW/m^2^ were compared with the CP (30°)-1 specimen temperature responses. Compared to the CP (30°)-1 specimen, the fabrication techniques used by Milos et al. are more complex. The comparison shows the woven carbon phenolic material specimen’s exposed surface temperature (i.e., the stagnation point temperature measured by a pyrometer) was higher than that of the CP (30°)-1 specimen. Though, the CP (30°)-1 specimen’s internal temperatures at 20 and 30 mm, were higher than the woven carbon phenolic material’s internal temperatures at 20.32 and 25.4 mm, the CP (30°)-1 specimen’s internal temperatures at 10 mm is comparable with the woven carbon phenolic material’s internal temperatures at 10.16 mm. Overall, considering the complexity of the NASA’s dual-layer woven carbon phenolic material, the temperature responses of the CP (30°)-1 specimen are reasonable and acceptable.

Paglia et al. [5] fabricated two carbon phenolic ablators, denominated as MF and RF; using two different types of carbon felts using typical fabrication techniques and tested their specimens at two heat flux test conditions: 6 and 13 MW/m^2^. Here, in Figure 33, the temperature responses from the specimens studied by Paglia et al. at 6 MW/m^2^ were compared with the CP (30°)-2 specimen’s temperature responses. The comparison shows the CP (30°)-2 specimen’s exposed surface temperature was lower than those of the compared ablators. The CP (30°)-2 specimen’s internal temperature at 30 mm was almost identical to the back face temperatures of the compared specimens; Paglia et al. specimens’ thickness was 40 mm.

Comparing recession rates, the CP (30°)-1 specimen’s stagnation point recession rate is 2.4-fold greater than the predicated stagnation point recession rate of the Milos et al. specimen; When compared to the Paglia et al. specimens, CP (30°)-2 specimen’s stagnation point recession rate is 0.54-fold lower and mass loss rate is 0.6-fold lower. Milos et al. did not publish their mass loss rate; hence a comparison with their mass loss rate is not made here. In other words, it can be said that the 30° carbon phenolic ablator’s recession and mass loss rates are better than the Paglia et al. specimens, which were fabricated using typical techniques (similar to the 30° carbon phenolic ablator) and further improvement required to match the recession rate of the Milos et al. specimen, which was fabricated using complex techniques.

Comparisons (Figure 32 and Figure 33 and Table 3) show that the 30° carbon phenolic ablator developed and tested in this study is on par with other internationally developed and studied carbon phenolic ablators. 

### Limitations

Here, specimens were tested at stationary positions (i.e., at fixed heat flux test conditions) but in real ballistic re-entries, heat flux subjected by spacecraft varies with time. Spacecraft’s heat flux is zero before the point of re-entry, then starts to increase after the point of re-entry. It peaks within the atmosphere and then it decreases and finally reaches zero before landing on the ground [29]. For more accurate simulation tests, the specimens may be tested transiently in the test flow to achieve heat flux test profiles similar to real re-entry heat flux trajectories. This limitation is currently being addressed, and mechanical changes to the HVOF material ablation facility are being made to facilitate transient heat flux tests.

Though material ablation tests using the HVOF test facility provide excellent and comprehensive material response results, the test flow is combustion based; the gas–material surface interactions occurred in these tests are not same as the real gas–material surface interactions that happen in an atmospheric re-entry. More accurate and closer to real gas–material surface interaction studies are possible only using plasma wind tunnels, as the atmospheric gases become plasma during a re-entry. Hence, this study can be considered as a preliminary investigation. More advanced tests using plasma wind tunnels will be planned in the future. 

Although the periodic explosions observed in the 0° carbon phenolic specimens are due to the lamination angle, the exact mechanism behind them needed to be confirmed. This requires further analysis using more sophisticated techniques, which are not within the scope of our current research work. Further analysis of this observed phenomenon will be planned and carried-out in our future research.

## 4. Conclusions

In this study, candidate materials for TPS application were developed and tested at several heat flux conditions consistent with a re-entry trajectory of a sample return space capsule. The specimens were tested at a HVOF material ablation test facility. Two types of test specimens were used. One is carbon phenolic ablators which have 0° and 30° lamination angles. The other is SiC-coated C–C specimens with an inner YSZ coating layer and with either a cork or a graphite base at the lower section.

The SiC-coated C–C specimens’ exposed surfaces demonstrated exceptional resistance to ablation and exhibited negligible levels of recession and mass loss. The average values of recession and mass loss are 0.14 mm/min and 0.7 g/min, respectively. However, the performances of their lower sections were poor. The graphite section’s internal temperatures values are the highest. While the 30° carbon phenolic specimen’s (CP (30°)-4) internal temperature values are approximately 1.5-fold lower at the heat flux of 11.5 MW/m^2^. The CP (30°)-4 specimen’s mass loss value is approximately 23-fold greater than that of the SiC/graphite specimen and the recession value of the CP (30°)-4 specimen is approximately 44-fold greater than that of the SiC/graphite specimen. The increased material ablation observed in the 30° carbon phenolic specimens is considered to be responsible for its lower internal temperatures when compared to the SiC/graphite specimen.

The 0° carbon phenolic material considered is inappropriate for TPS applications. The reason is that during the 0° carbon phenolic specimen tests, periodic explosions were observed. Further investigation of the explosion phenomenon is required. 

Considering the overall results, the 30° carbon phenolic specimens are the most reliable and eligible for TPS application among the tested specimens, with the main factor being the lower internal temperature. In addition, the 30° carbon phenolic materials exhibit a normal ablation reaction, unlike the 0° carbon phenolic materials.

By comparing the pyrometer and IR camera data, the estimated emissivity values were found to be 0.85 ± 0.02 for carbon phenolic specimens and 0.87 ± 0.01 for SiC-coated C–C specimens.

Future work will focus on further developing and vigorously testing the carbon phenolic material using the advanced ground test facilities such as arc-jet or inductively coupled plasma wind tunnels.

## Figures and Tables

**Figure 1 materials-16-01895-f001:**
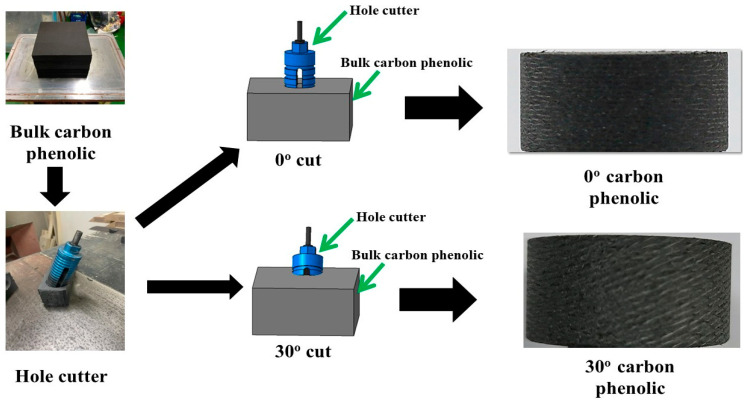
Carbon phenolic specimen fabrication process.

**Figure 2 materials-16-01895-f002:**
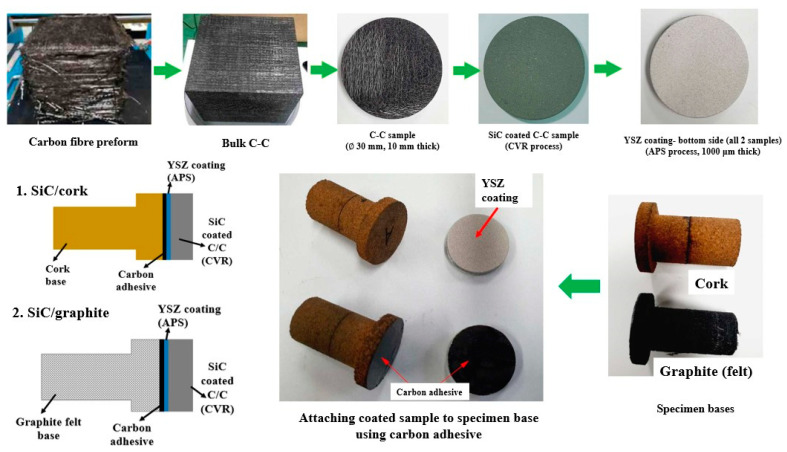
SiC-coated C–C specimen fabrication process.

**Figure 3 materials-16-01895-f003:**
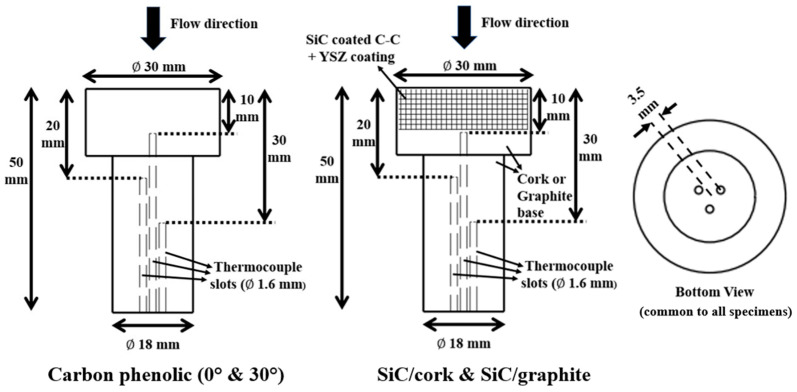
Specimen dimensions.

**Figure 4 materials-16-01895-f004:**
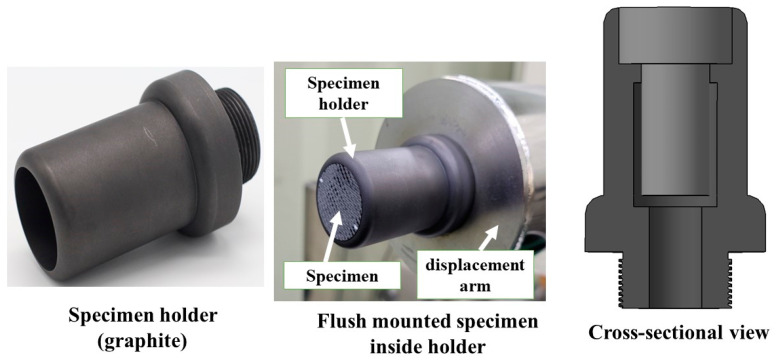
Specimen holder.

**Figure 5 materials-16-01895-f005:**
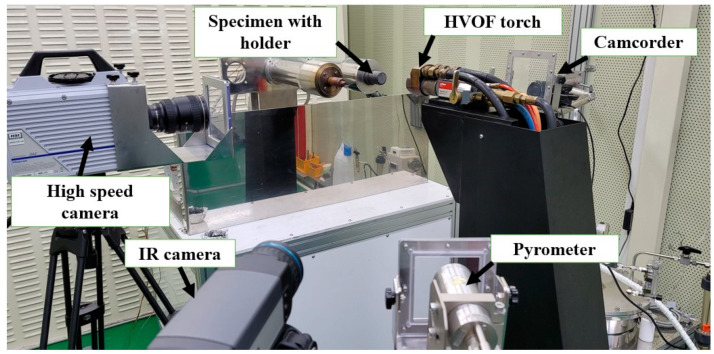
Experimental setup photograph.

**Figure 6 materials-16-01895-f006:**
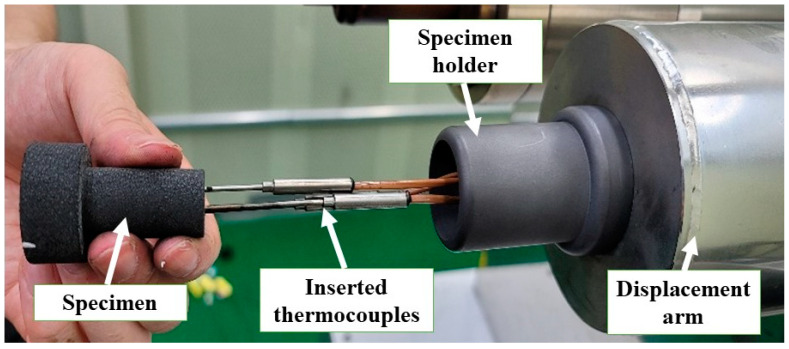
Specimen with inserted thermocouples.

**Figure 7 materials-16-01895-f007:**
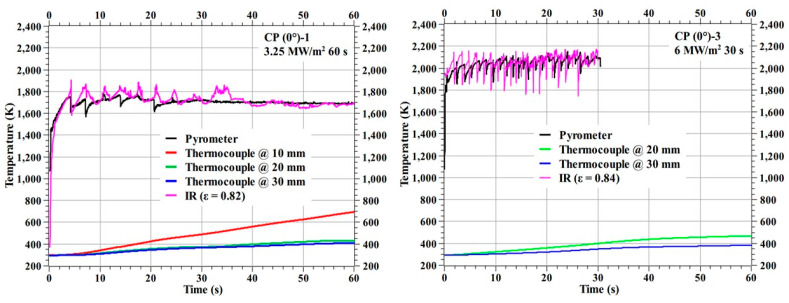
CP (0°)-1 and CP (0°)-3 temperature responses.

**Figure 8 materials-16-01895-f008:**
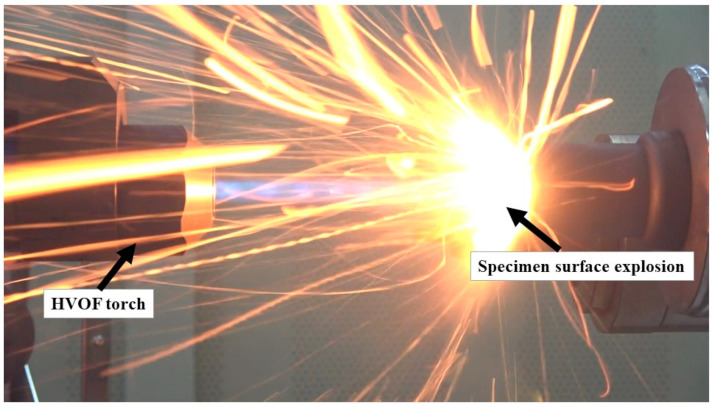
Explosion on CP (0°)-3 specimen surface.

**Figure 9 materials-16-01895-f009:**
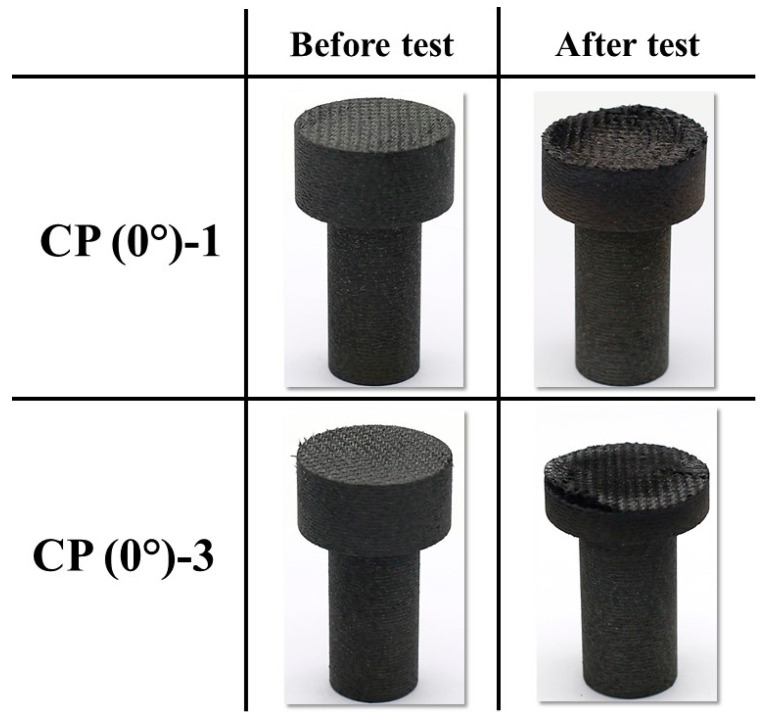
CP (0°)-1 (3.25 MW/m^2^, 60 s) and CP (0°)-3 (6 MW/m^2^, 30 s) before and after test images.

**Figure 10 materials-16-01895-f010:**
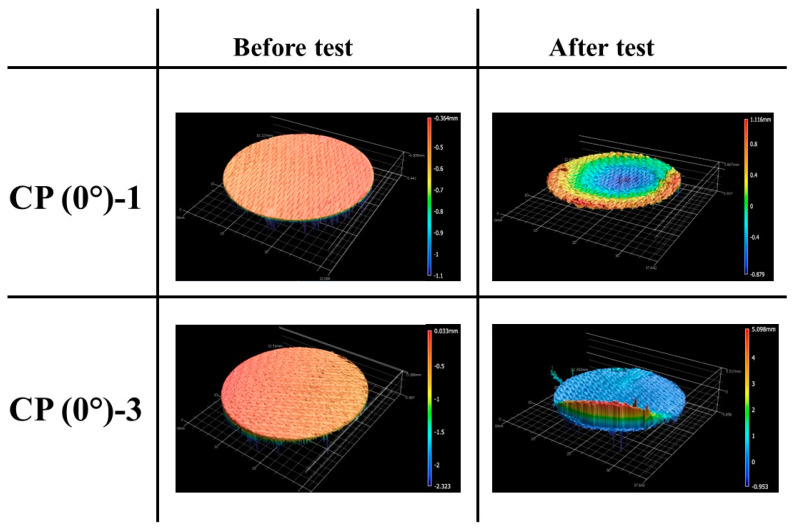
CP (0°)-1 (3.25 MW/m^2^, 60 s) and CP (0°)-3 (6 MW/m^2^, 30 s) exposed surface morphology changes.

**Figure 11 materials-16-01895-f011:**
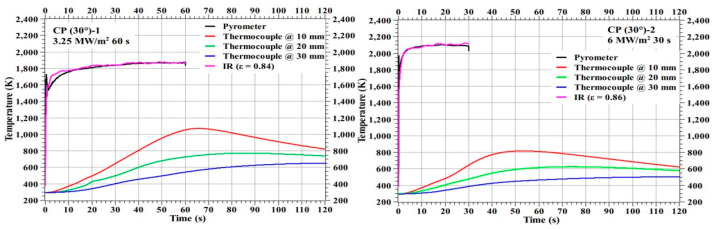
CP (30°)-1 and CP (30°)-2 temperature responses.

**Figure 12 materials-16-01895-f012:**
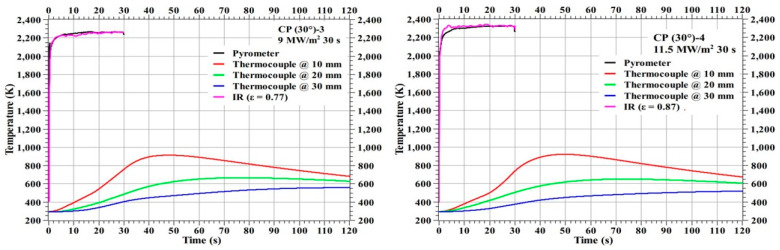
CP (30°)-3 and CP (30°)-4 temperature responses.

**Figure 13 materials-16-01895-f013:**
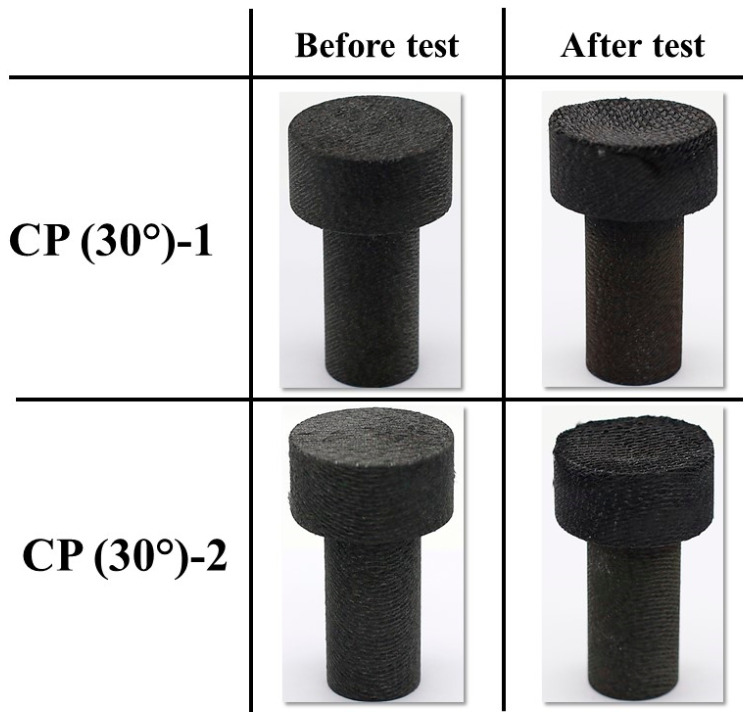
CP (30°)-1 (3.25 MW/m^2^, 60 s) and CP (30°)-2 (6 MW/m^2^, 30 s) before and after test images.

**Figure 14 materials-16-01895-f014:**
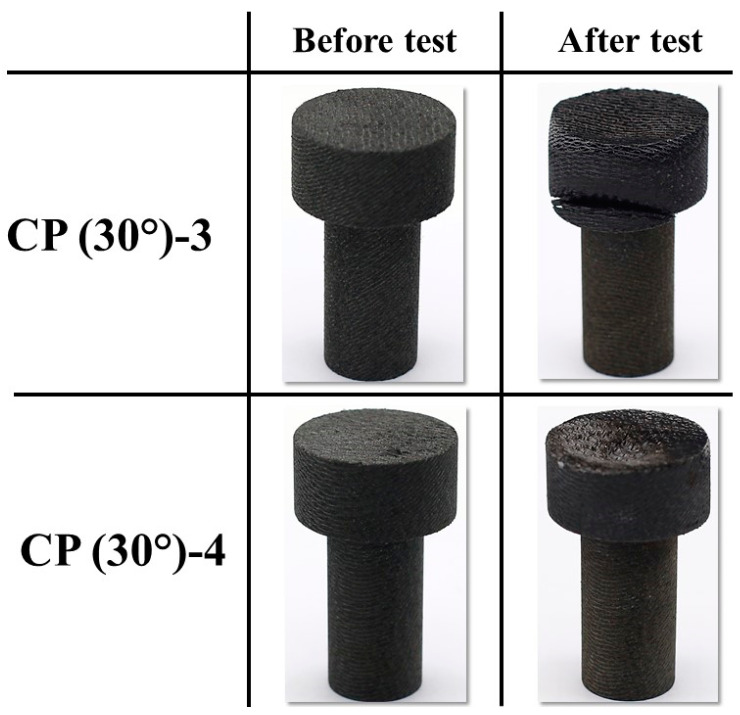
CP (30°)-3 (9 MW/m^2^, 30 s) and CP (30°)-4 (11.5 MW/m^2^, 30 s) before and after test images.

**Figure 15 materials-16-01895-f015:**
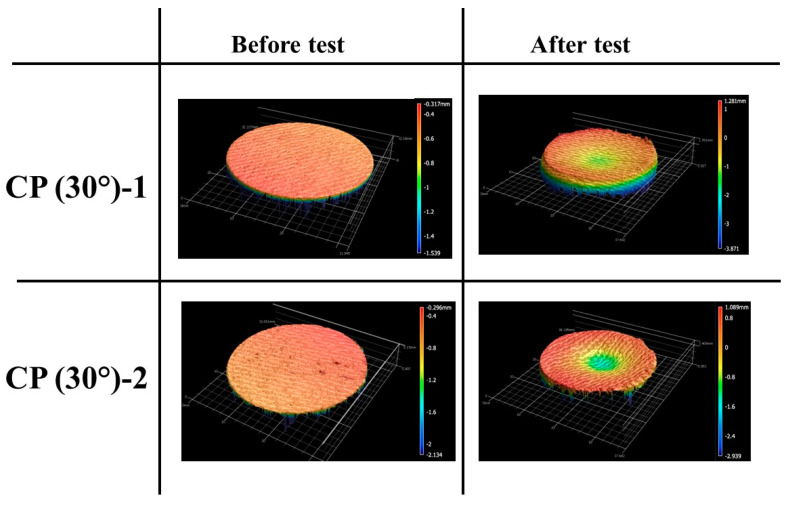
CP (30°)-1 (3.25 MW/m^2^, 60 s) and CP (30°)-2 (6 MW/m^2^, 30 s) exposed surface morphology changes.

**Figure 16 materials-16-01895-f016:**
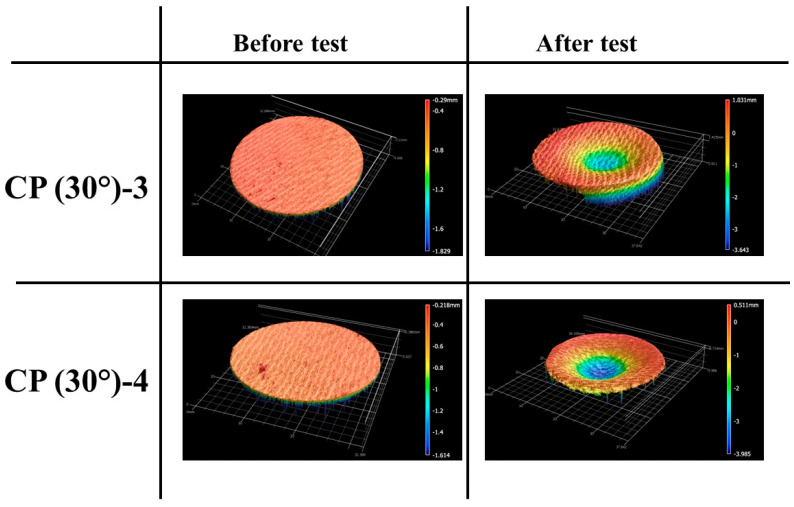
CP (30°)-3 (9 MW/m^2^, 30 s) and CP (30°)-4 (11.5 MW/m^2^, 30 s) exposed surface morphology changes.

**Figure 17 materials-16-01895-f017:**
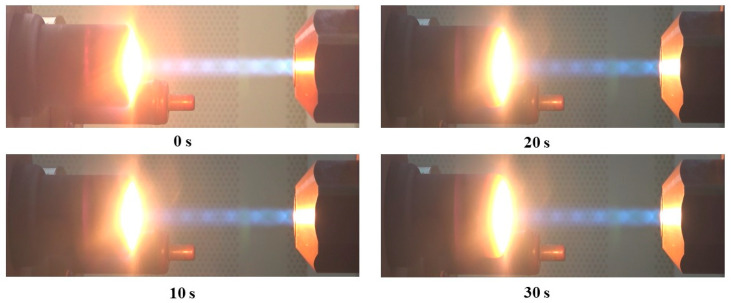
Camcorder images of the CP (30°)-4 specimen.

**Figure 18 materials-16-01895-f018:**
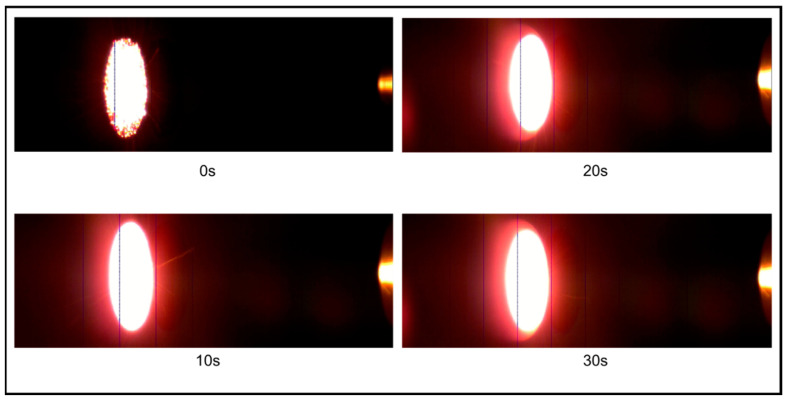
High-speed camera images of the CP (30°)-4 specimen.

**Figure 19 materials-16-01895-f019:**
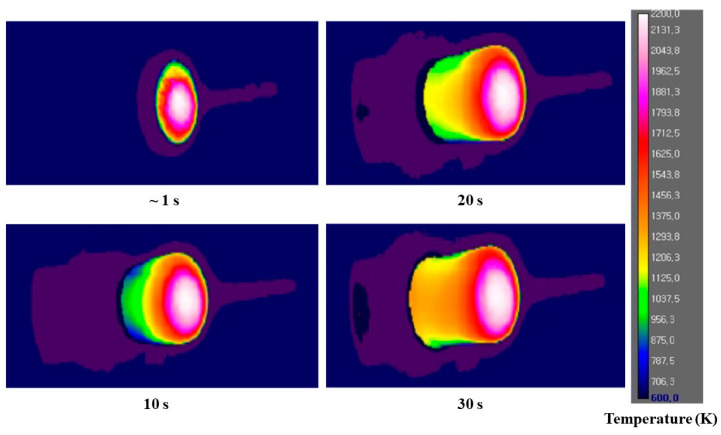
IR images of the CP (30°)-4 specimen.

**Figure 20 materials-16-01895-f020:**
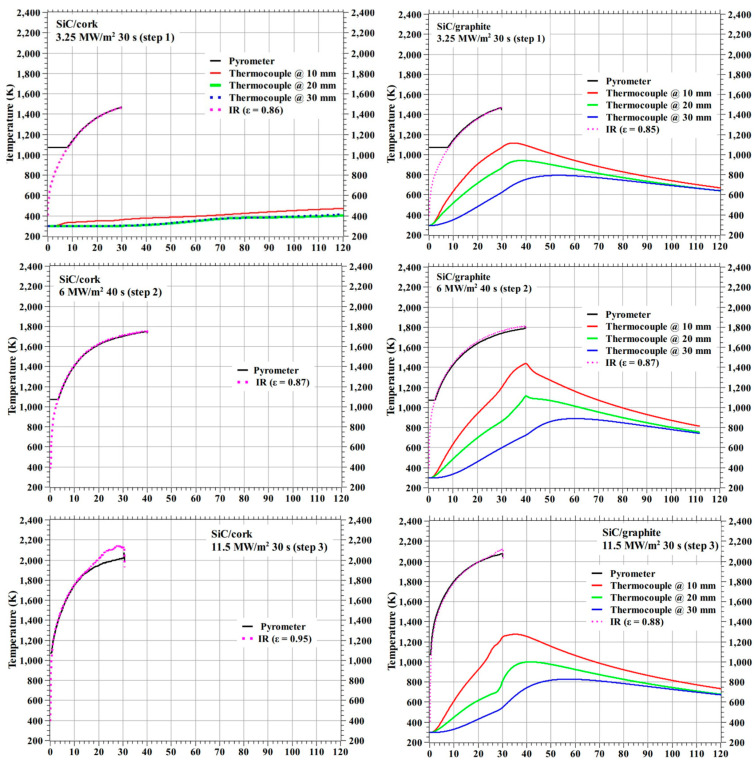
SiC-coated C–C specimens’ temperature responses.

**Figure 21 materials-16-01895-f021:**
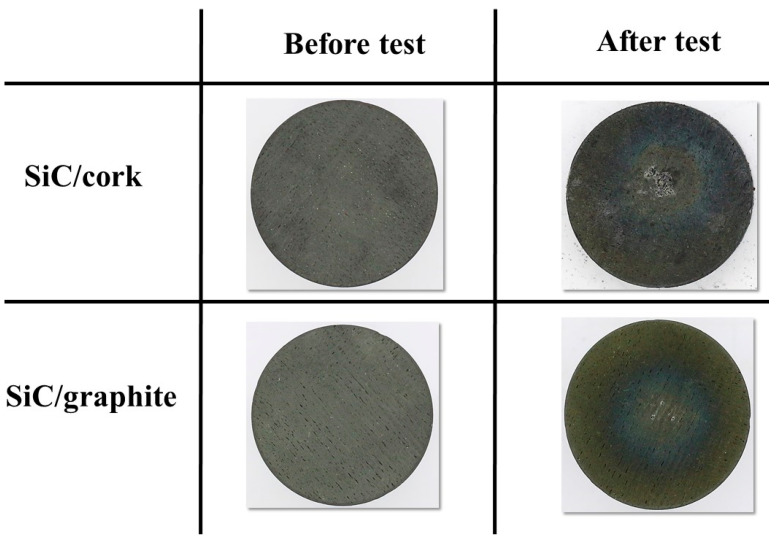
SiC-coated C–C specimens’ before and after test images (exposed surface).

**Figure 22 materials-16-01895-f022:**
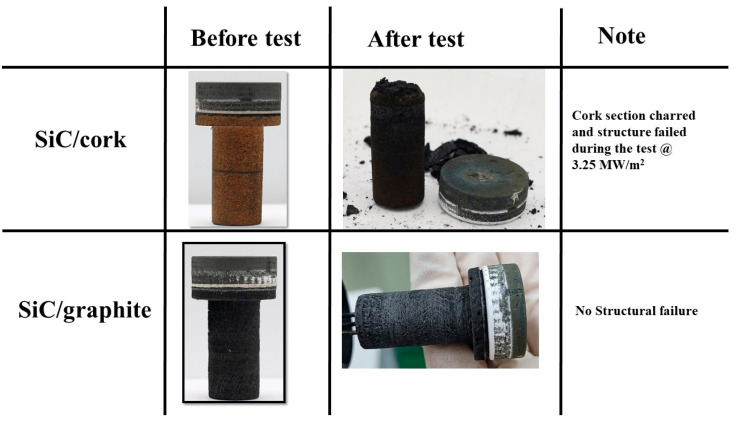
SiC-coated C–C specimens’ before and after test images (side view).

**Figure 23 materials-16-01895-f023:**
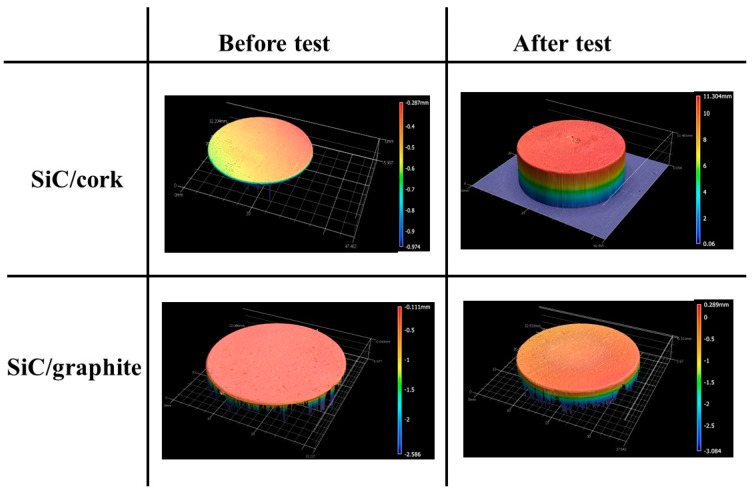
SiC-coated C–C specimens’ exposed surface morphology changes.

**Figure 24 materials-16-01895-f024:**
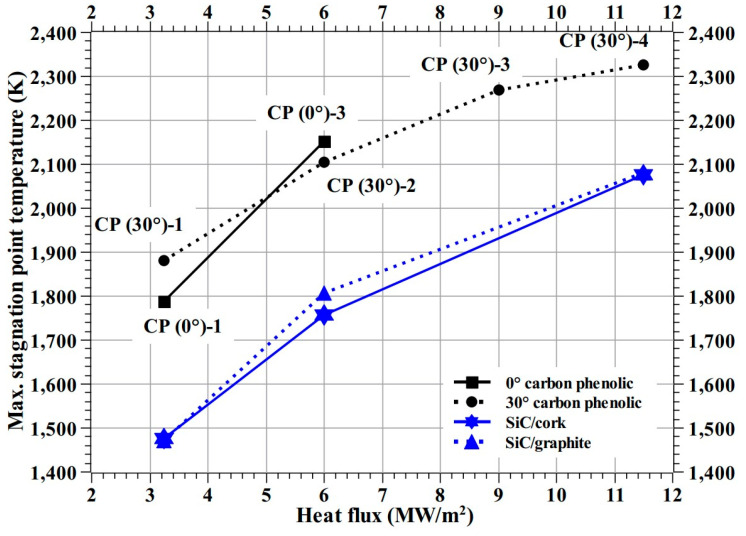
Maximum stagnation point temperature vs. heat flux.

**Figure 25 materials-16-01895-f025:**
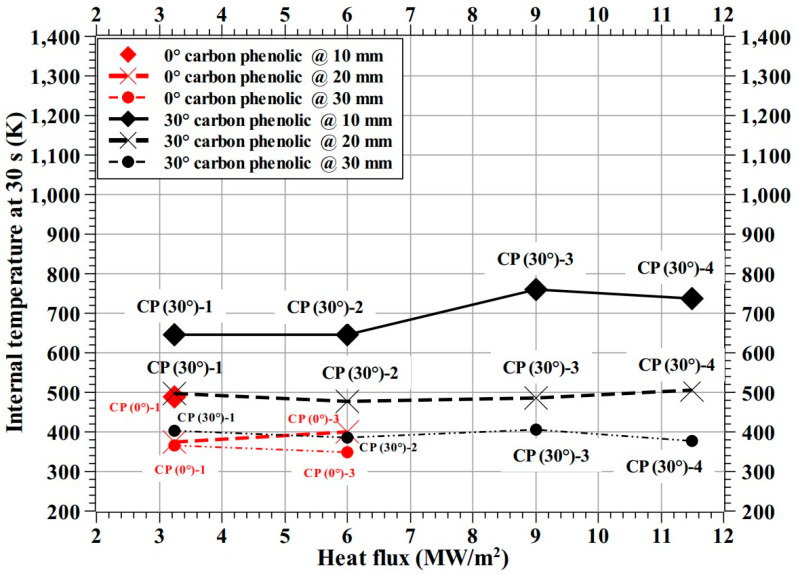
Internal temperature at 30 s (carbon phenolic 0° and 30°).

**Figure 26 materials-16-01895-f026:**
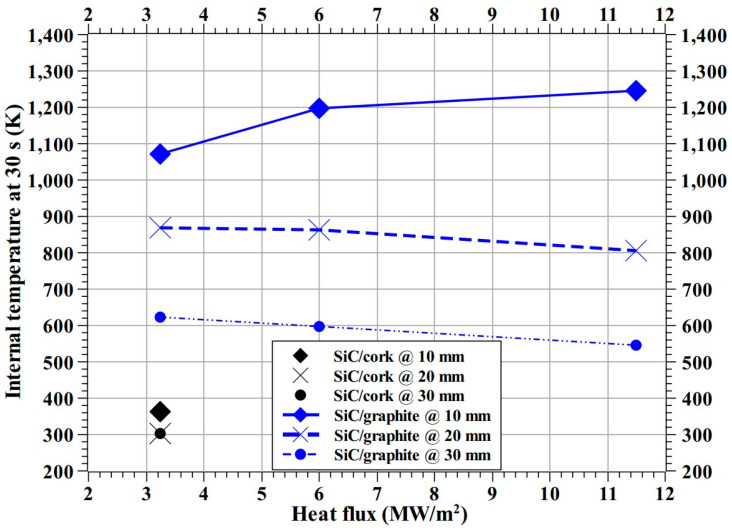
Internal temperature at 30 s (SiC/cork and SiC/graphite).

**Figure 27 materials-16-01895-f027:**
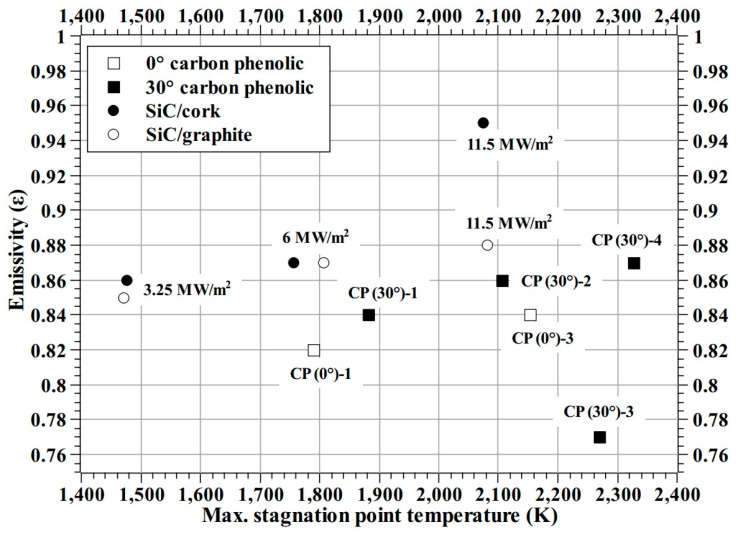
Test wise estimated emissivity (ε).

**Figure 28 materials-16-01895-f028:**
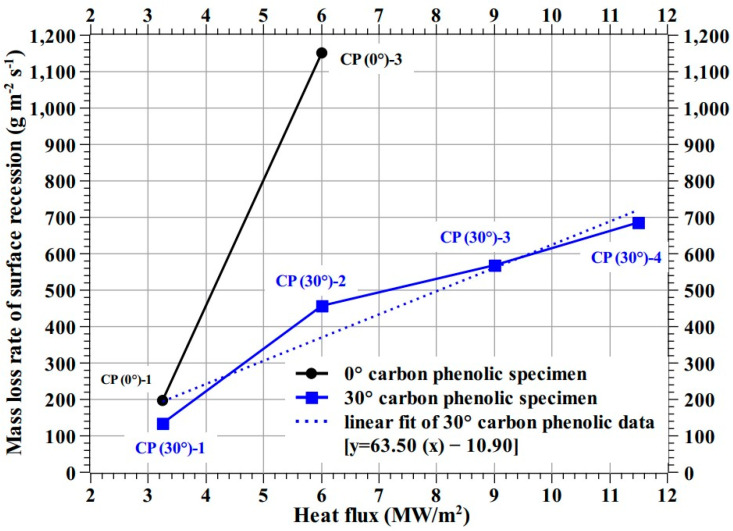
Mass loss rate of surface recession (carbon phenolic).

**Figure 29 materials-16-01895-f029:**
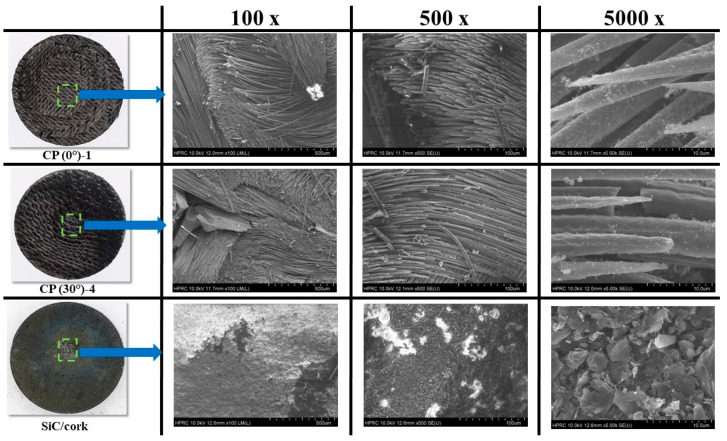
Specimen exposed surface SEM images.

**Figure 30 materials-16-01895-f030:**
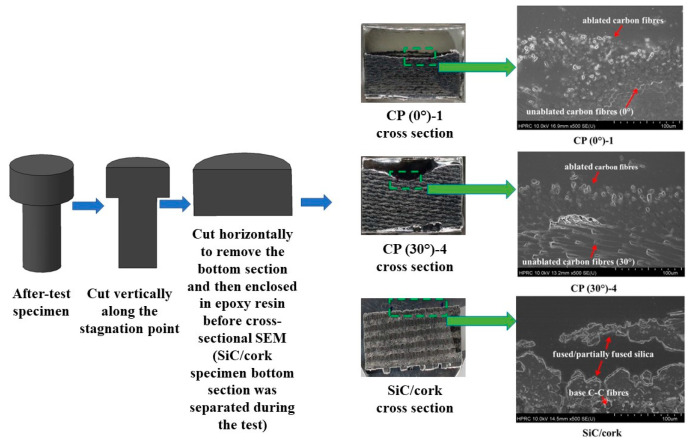
Specimen cross-sectional SEM images (500×).

**Figure 31 materials-16-01895-f031:**
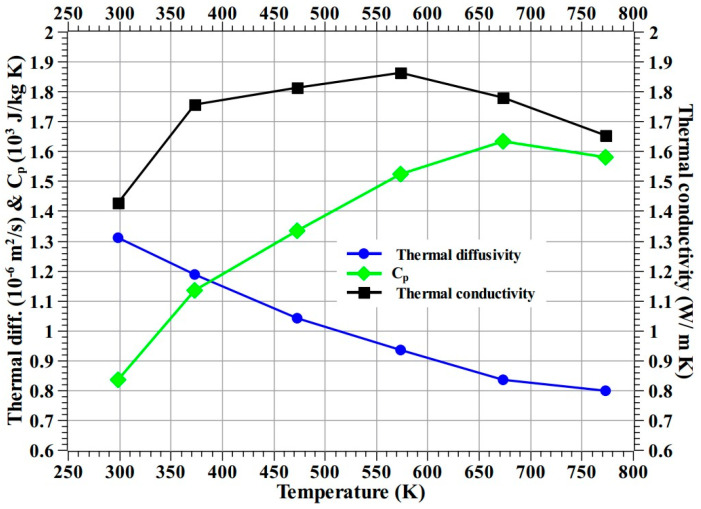
Thermophysical properties of 30° carbon phenolic up to 773.15 K (500 °C).

**Figure 32 materials-16-01895-f032:**
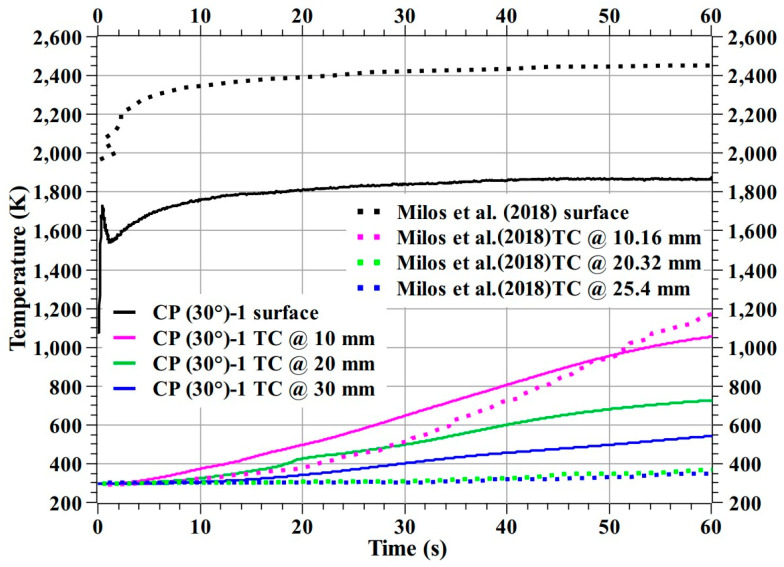
Comparison of CP (30°)-1 and NASA’s dual-layer woven carbon phenolic temperature responses [28].

**Figure 33 materials-16-01895-f033:**
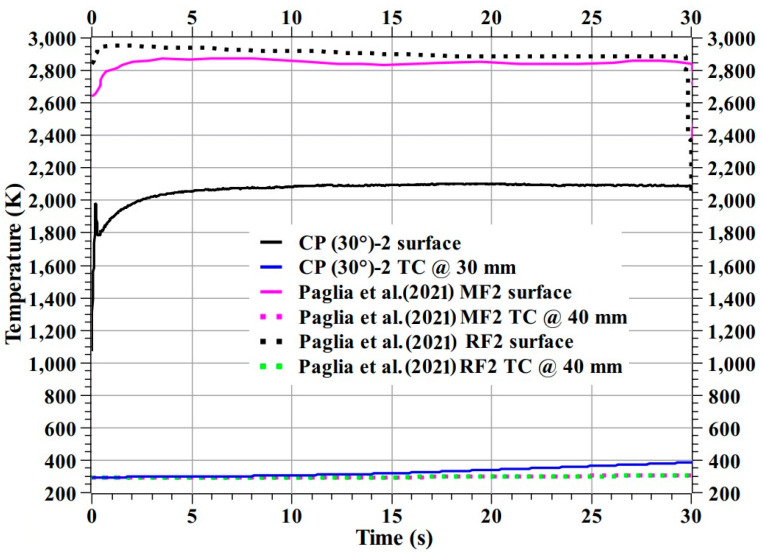
Comparison of CP (30°)-2 and Paglia et al. carbon phenolic ablators temperature responses [5].

**Table 1 materials-16-01895-t001:** Specimen test conditions.

No.	Specimen	Heat Flux (MW/m^2^)	Test Duration (s)	Distance from the HVOF Nozzle Exit (mm)
1	CP (0°)-1	3.25	60	150
2	CP (0°)-3	6	30	120
3	CP (30°)-1	3.25	60	150
4	CP (30°)-2	6	30	120
5	CP (30°)-3	9	30	100
6	CP (30°)-4	11.5	30	90
7	SiC/cork	3.25	30	150
		6	40	120
		11.5	30	90
	SiC/graphite	3.25	30	150
8		6	40	120
		11.5	30	90

CP (0°) = carbon phenolic (0° lamination). CP (30°) = carbon phenolic (30° lamination). SiC/Cork = SiC-coated C–C + cork base. SiC/graphite = SiC-coated C–C + graphite felt base.

**Table 2 materials-16-01895-t002:** Specimen mass loss rate and recession rate.

No.	Specimen	Mass Loss Rate (g/min)	Recession Rate (mm/min)
1	CP (0°)-1	5.1	5.28
2	CP (0°)-3	9.96	7.02
3	CP (30°)-1	4.42	3.63
4	CP (30°)-2	6.58	6.14
5	CP (30°)-3	7.34	7.64
6	CP (30°)-4	7.68	9.2
7	SiC/cork	1.06	0.06
8	SiC/graphite	0.34	0.21

Note: For the SiC-coated C–C specimens, mass loss and recession were estimated after the final test, i.e., after the third step at 11.5 MW/m^2^.

**Table 3 materials-16-01895-t003:** Comparison of mass loss rates and recession rates.

Specimen	Heat Flux (MW/m^2^)	Recession Rate (mm/min)	Mass Loss Rate (g/min)
CP (30°)-1	3.25	3.63	4.42
Milos et al. [28]		1.5	-
CP (30°)-2	6	6.14	6.58
Paglia et al. RF2 [5]		11.28	11.30
Paglia et al. MF2 [5]		11.34	11.56

## Data Availability

The data will be made available on request from the corresponding author.

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
