# Peer review of "Thermal Ablation Experiments of Carbon Phenolic and SiC-Coated Carbon Composite Materials Using a High-Velocity Oxygen-Fuel Torch"

_materials, 2023, doi:10.3390/ma16051895_

Round 1
Reviewer 1 Report
Title: Thermal Ablation Experiments of Carbon Phenolic and SiC 2 Coated Carbon Composites Materials using a High Velocity 3 Oxygen-Fuel Torch
In this work, ablation experiments of carbon phenolic material specimens with two lamination angles (0° and 30°) and two specially designed SiC coated carbon composite specimens were conducted using an HVOF torch ablation test facility. The heat flux test conditions ranged from 3.25 MW/m2 to 11.5 MW/m2, corresponding to an interplanetary sample return re-entry heat flux trajectory. A two-color pyrometer, an IR camera, and thermocouples (three internal locations) were used to measure the specimen temperature responses. The temperature responses of the 30° carbon phenolic material were comparable with other carbon phenolic ablators.
I, recommend this paper be published in the Materials Journal after the authors address the following comments.
· Review English grammar as there are mistakes throughout the text.
· The abstract must be written quantitatively and focused on the findings.
· Please add the nomenclature.
· The physical explanation of figures 23-31 is limited. Please explain more.
The author must improve the introduction with more advanced applications. Also, the author could find new references for the literature review. Such as:
v Materials Science and Engineering: B 274 (2021): 115458.
v Materials 15.16 (2022): 5695.
v Graphene and 2D Materials
· The conclusion is too long and so weak. Please explain your main findings quantitatively.
· The literature section must be improved with more advanced articles and clearly why your present study is different, better to explain novelty. The innovation of the study should be presented in the introduction part.
· Add a section on study limitations before the conclusion section.
· L:214: On the contrary, for the 0° carbon phenolic specimens, due to periodic explosions and layer-by-layer removal of laminate layers, the material loss occurred throughout the surface perpendicular to the test flow direction. Why?
· The number of figures is very high. Merge the figures.
· L: 370: The comparison shows 370 the CP (30°)-2 specimen’s exposed surface temperature was lower than those of the com-371 pared ablators. Why?
In conclusion, this paper might be made suitable for publication in this Journal if the as-mentioned comments are clarified. These constitute a Major revision of it.
Author Response
Thank you for your valuable time and comments.
Please see the attached document and revised paper.

Reviewer 2 Report
Referee report on “Thermal Ablation Experiments of Carbon Phenolic and SiC Coated Carbon Composites Materials using a High Velocity Oxygen-Fuel Torch”
Although this topic is of some interest, this manuscript in its present form cannot be recommended for publication and requires some improvement and clarification.
1. A clear disadvantage is the lack of understanding and specifics and what crystalline modification of silicon carbide is meant.
2. Furthermore, the introduction needs more general information about SiC and its important applications in optical devices, nanotechnology and nuclear and space material science. This is important to attract more reader interest and further incentive applications. For some of them, see, for example:
a) Huczko, A., Dąbrowska, A., et al . Silicon carbide nanowires: synthesis and cathodoluminescence. physica status solidi (b), 2009, 246(11‐12), 2806-2808.
b) Jalluri, T. D., Gouda, G. M., Dey, A., Rudraswamy, B., & Sriram, K. V. (2022). Development and characterization of silicon dioxide clad silicon carbide optics for terrestrial and space applications. Ceramics International, 48(1), 96-110.
c) Ning G., Zhang L., Zhong W., Wang S., Liu J., Zhang C. Damage and annealing behavior in neutron-irradiated SiC used as a post-irradiation temperature monitor
(2022) Nuclear Instruments and Methods in Physics Research, Section B: Beam Interactions with Materials and Atoms, 512, pp. 91-95.
3. The introduction does not reflect the history of the research data, its relevance and prospects. References are all quite old and this only confirms what has been said.
In this form, the manuscript contains more engineering data than material characteristics. Therefore, it seems to me that this article is more suitable for a journal like Applied Sciences than Materials.
Therefore, in the conclusions, it is necessary to clearly formulate what new data about the studied materials were obtained in this work?
In general, the manuscript is interesting and can be considered for publication after constructive reflection on the above comments.
Author Response

(The authors gave the same response as above.)

Reviewer 3 Report
good and very clear paper on ablation of C/C composites. The effect of fibres orientations is very well decribed.
Some minor remarks may be done:
- in table 1: specimens n° 7 and 8 are not well described: 3.25/6/11.5 shall be replaced respctivley by: SiC/Cork and SiC/graphite as mentionned below the table.
- in the captions of fig 15/16/17/18 remind the levels of the heat flux for each specimens as explained in the table 1.
- fig 30 is very interesting: why SiC/Cork and SiC/graphite are not plotted on this figure? for the calculation of equation (2) is the couple (heat/flux = 0 ; mass loss rate = 0) introduced in the fitting? why a linear fitting has been chosen, the intercept '-10.90' is really significant? a polynomial fitting (order 2 or 3) would not be prefered?
- at the end of the paper is it possible to compare also the mass loss rate and/or the recession rate of CP (30°) with Milos and Paglia composites? are these rates better for CP (30°) than for Milos and Paglia composites whatever the level of the heat flux?
- at the end of the paper a short comment shall be added also to well explain why CP (0°) exploxes and not CP(30°). is it due to the fact that fibres at 30° act as a mechanical reinforcement with fibre/matrix interactions, or fibres allow a better conduction of the heat locally and a better homogeneity of the temperature field?
the outlooks at the end of the conclusion section have been well appreciated.
Author Response

(The authors gave the same response as above.)

Round 2
Reviewer 1 Report
Accept
Reviewer 2 Report
After successful revision, this paper can be accepted.